# A novel Cep120-dependent mechanism inhibits centriole maturation in quiescent cells

**Ewelina Betleja[1], Rashmi Nanjundappa[1], Tao Cheng[1], Moe R Mahjoub[1,2]***

[1]Department of Medicine (Nephrology Division), Washington University, St Louis, United States; [2]Department of Cell Biology and Physiology, Washington University, St Louis, United States

**Abstract** The two centrioles of the centrosome in quiescent cells are inherently asymmetric structures that differ in age, morphology and function. How these asymmetric properties are established and maintained during quiescence remains unknown. Here, we show that a daughter centriole-associated ciliopathy protein, Cep120, plays a critical inhibitory role at daughter centrioles. Depletion of Cep120 in quiescent mouse and human cells causes accumulation of pericentriolar material (PCM) components including pericentrin, Cdk5Rap2, ninein and Cep170. The elevated PCM levels result in increased microtubule-nucleation activity at the centrosome. Consequently, loss of Cep120 leads to aberrant dynein-dependent trafficking of centrosomal proteins, dispersal of centriolar satellites, and defective ciliary assembly and signaling. Our results indicate that Cep120 helps to maintain centrosome homeostasis by inhibiting untimely maturation of the daughter centriole, and defines a potentially new molecular defect underlying the pathogenesis of ciliopathies such as Jeune Asphyxiating Thoracic Dystrophy and Joubert syndrome.

DOI: https://doi.org/10.7554/eLife.35439.001

*For correspondence:
mmahjoub@wustl.edu

**Competing interests:** The authors declare that no competing interests exist.

## Introduction

The centrosome is the major microtubule-organizing center (MTOC) in animal cells, which helps regulate the assembly of the interphase microtubule array and mitotic spindle. The centrosome in a quiescent cell is composed of a pair of centrioles surrounded by a proteinaceous matrix referred to as pericentriolar material (PCM), involved in microtubule nucleation and anchoring. The two centrioles in quiescent cells differ in age, structure and function. The older of the two was assembled at least two cell cycles ago and is commonly called the 'mother' centriole (MC). The younger one, which was assembled in the previous cell cycle, is called the 'daughter' centriole (DC). Most activities at the centrosome are regulated and mediated by the mother centriole. For example, the MC acts as the template for the assembly of the primary cilium, an essential chemosensory organelle (*Malicki and Johnson, 2017*). The MC possesses distinct sets of projections called distal appendages that confer the ability to dock at the plasma membrane and initiate ciliogenesis, and subdistal appendages that help organize the PCM, nucleate and anchor microtubules (*Garcia and Reiter, 2016*). In contrast, the DC lacks distal and subdistal appendages, does not form a cilium, and possesses significantly less PCM and thus MTOC activity. How and why the DC is prohibited from recruiting these protein complexes remains a mystery. The predominant theory is that the DC undergoes structural and biochemical modifications during the cell cycle that confers the ability to acquire these traits, a multi-step process classically referred to as 'maturation' (*Hoyer-Fender, 2010*). However, recent studies suggest that this model may be too simplistic (*Kong et al., 2014*; *Wang et al., 2011*). So, how is the DC prohibited from recruiting these protein complexes during quiescence?

**eLife digest** Among the countless components of an animal cell, microtubules perform many important roles. These hollow filaments support the cell's shape and help to transport different materials around within it. They also form a hair-like projection on the cell surface called the primary cilium, which helps the cell sense its environment. Most microtubules in an animal cell are organized by a structure called the centrosome, which has two smaller cylindrical structures called centrioles at its core. In cells that are not dividing, these two centrioles are different in age. The older of the two centrioles was assembled at least two cell divisions ago and is commonly called the "mother" centriole. The younger one, which was assembled the previous time the cell divided, is called the "daughter" centriole.

Most activities at the centrosome are controlled by the mother centriole. For example, the mother centriole contains protein complexes called appendages that allow it to dock at the cell surface and build the cilium. The mother centriole also contains a complex of proteins called the pericentriolar material, which helps it assemble microtubules and anchor them in place. In contrast, the daughter centriole lacks appendages, does not form a cilium, has less pericentriolar material and so assembles fewer microtubules. Why the daughter centriole cannot recruit these protein complexes remains a mystery.

One possibly important difference between mother and daughter centrioles is that daughter centrioles in non-dividing cells have much higher levels of a protein called Cep120. Now, Betleja et al. have studied the role of this protein in more detail. Experiments with mouse and human cells show that Cep120 plays an important inhibitory role at the daughter centriole. When the production of Cep120 was blocked, more pericentriolar material associated with the daughter centriole, and more microtubules were assembled by the centrosome. This interfered with the movement of other proteins to the centrosome, which ultimately disrupted both the centrosome's ability to assemble cilia and the cell's ability to sense its environment.

The findings of Betleja et al. show that a Cep120-dependent mechanism actively regulates the centrosome's function in non-dividing cells. These experiments uncover a potentially new type of molecular defect that may be responsible for diseases caused by faulty cilia, such as Joubert Syndrome and Jeune Asphyxiating Thoracic Dystrophy. The next challenge will be to understand how Cep120 inhibits the levels of pericentriolar material only at the daughter centriole but not the mother centriole.

DOI: https://doi.org/10.7554/eLife.35439.002

We previously identified Cep120 as a centrosomal protein that is preferentially enriched on the DC of quiescent cells (*Mahjoub et al., 2010*). This asymmetric localization between the mother and daughter centriole was relieved coincident with new procentriole assembly in S-phase, where Cep120 became enriched on the procentrioles ([*Mahjoub et al., 2010*]; summarized in *Figure 1A*). Thus, Cep120 associates with the youngest generation of centrioles at all stages of the cell cycle, although some of the protein remains on the MC ([*Mahjoub et al., 2010*]; *Figure 1A*). To determine the functional significance of Cep120 enrichment on procentrioles, we depleted the protein and analyzed the consequences on centriole assembly. We found that Cep120 was required for procentriole formation in S-phase of cycling cells, and for nascent centriole assembly in multiciliated cells (*Mahjoub et al., 2010*), revealing a conserved role for Cep120 in centriole formation. Subsequent studies helped to define the role of Cep120 in the molecular pathways regulating procentriole assembly (*Comartin et al., 2013*; *Lin et al., 2013*). More recently, mutations in Cep120 were identified in patients with ciliopathy phenotypes, namely Joubert syndrome and Jeune Asphyxiating Thoracic Dystrophy (*Roosing et al., 2016*; *Shaheen et al., 2015*), further highlighting a critical role in centriole-centrosome-cilia biology. However, one major question remains unanswered: what is the functional significance of the enrichment of Cep120 on the DC of quiescent cells, at a stage of the cell cycle when centriole duplication is not occurring?

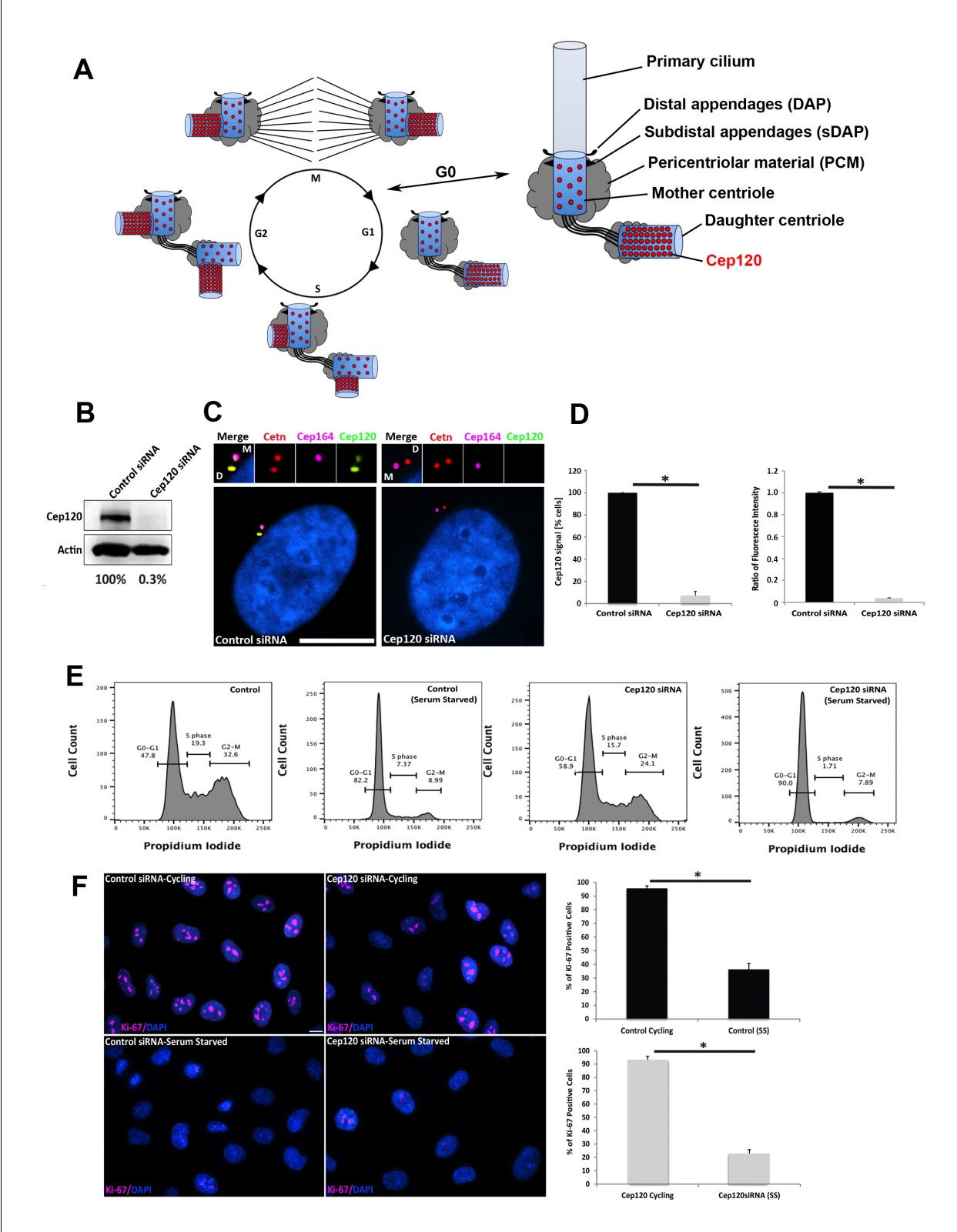

**Figure 1.** Depletion of Cep120 in quiescent cells. (A) Schematic of centriole duplication and the cell cycle. Cep120 is enriched on the youngest generation of centrioles at all stages of cell division. Importantly, Cep120 is asymmetrically localized and enriched on the daughter centriole during $G_0$. (B) Cep120 depletion from MEF cells by siRNA transfection. Lysates were probed for Cep120 and actin (loading control). Numbers below indicate relative levels of Cep120, normalized to actin. (C) MEFs were transfected with the indicated siRNA, serum starved for 24 hr, fixed, and stained for

*Figure 1 continued on next page*

*Figure 1 continued*

Cep120, centrin (centrioles), Cep164 (mother centriole) and DAPI (DNA). (D) (Left) Quantification of the fraction of transfected cells with Cep120 staining at the centrosome. N = 300 (control) and 300 (Cep120) siRNA. (Right) Ratio of fluorescence intensity of centrosomal Cep120, in control versus Cep120-depleted cells. N = 214 (control) and 244 (Cep120) siRNA. (E) FACS analysis performed on MEF cells transfected with control or Cep120 siRNA, incubated in normal growth medium for 24 hr, then in low-serum medium for another 24 hr. The percentage of cells at each cell cycle phase is indicated. (F) siRNA transfected MEF were grown for 24 hr in normal growth medium (cycling), then incubated in low-serum medium (serum-starved) for 24 hr. Cells were fixed and stained with antibodies against Ki-67 to identify proliferating cells. DNA was stained with DAPI. Graphs denote the fraction of Ki-67-positive cells during cycling and serum-starvation (SS) conditions. N = 300 (control cycling), N = 300 (control SS), N = 300 (Cep120 cycling), N = 300 (Cep120 SS). All results are averages of three independent experiments; *p<0.05. Scale bars = 10 μm.

DOI: https://doi.org/10.7554/eLife.35439.003

The following figure supplement is available for figure 1:

**Figure supplement 1.** Characterization of Cep120 antibody.

DOI: https://doi.org/10.7554/eLife.35439.004

## Results

### Depletion of Cep120 in quiescent cells

To address this question, we established an assay whereby we could deplete Cep120 while simultaneously arresting cells in $G_0$ and maintaining them in a quiescent state. This is critical because loss of Cep120 disrupts procentriole assembly in S-phase, resulting in cells that lack centrioles after a few cell divisions (*Mahjoub et al., 2010*). Thus, we could not utilize genome-editing approaches such as the CRISPR/Cas9 system, which entails isolation and screening of individual clones of cells, an approach that requires multiple passages and would thus disrupt centriole duplication in the absence of Cep120. To overcome this issue, we made use of siRNA oligos that are known to specifically and robustly deplete Cep120 (*Comartin et al., 2013*). Mouse embryonic fibroblasts (MEF) were seeded at high density, transfected with either control (non-targeting) or Cep120-specific siRNA, and immediately incubated in low serum-containing medium for 24–48 hr to arrest cells in $G_0$. Immunoblot analysis of whole-cell lysates showed this approach results in near complete depletion of Cep120 (*Figure 1B*). Consistent with this, there was loss of endogenous Cep120 signal from the DC, as evidenced by immunofluorescence staining of fixed cells (*Figure 1C–D*). Next, we analyzed the proportion of cells in each phase of the cell cycle by FACS, to ensure that Cep120-depleted cells are indeed arrested in $G_0$. Both control and Cep120-depleted MEF showed an increase in the $G_0$-$G_1$ population upon serum-starvation, with no significant difference in cell cycle profiles otherwise (*Figure 1E*). Finally, we immunostained cycling and serum-starved cells for Ki-67, a marker of cell proliferation expressed at all cell-cycle stages except $G_0$ (*Sobecki et al., 2017*). Both control and Cep120-depleted cells showed loss of Ki-67 signal upon serum removal (*Figure 1F*). Collectively, these data indicate that our experimental scheme results in robust depletion of Cep120 in quiescent cells, which still maintain their original mother and daughter centrioles.

### Loss of Cep120 in quiescent cells causes accumulation of PCM

We measured the abundance of asymmetrically localized proteins at the centrosome using quantitative single-cell immunofluorescence microscopy (described in detail in Methods), beginning with components of the PCM. Pericentrin is a core scaffolding protein that attaches to the centriolar microtubules and forms the foundation upon which other PCM components are assembled (*Doxsey et al., 1994*; *Mennella et al., 2014*). In MEF cells transfected with control siRNA, pericentrin showed preferential enrichment on the MC (*Figure 2A*). Depletion of Cep120 resulted in a roughly 2.5-fold increase in total centrosomal pericentrin (*Figure 2A–B*). Quantification of the relative levels of pericentrin on mother versus daughter centrioles showed that most of this increase occurred on the DC (*Figure 2B*). Cdk5Rap2 is recruited by pericentrin to help form the PCM matrix (*Fong et al., 2008*; *Mennella et al., 2014*), and is similarly enriched on the MC (*Figure 2C*). Loss of Cep120 caused a near 6-fold increase in total Cdk5Rap2 levels at the centrosome, with the majority of the increase seen at the DC (*Figure 2C–D*). Ninein is a PCM protein that localizes to the proximal ends of centrioles in addition to the subdistal appendages of the MC (*Bouckson-Castaing et al., 1996*; *Mazo et al., 2016*), and is thus enriched on the MC (*Figure 2E*). Cep120 depletion resulted in greater than 4-fold increase in centrosomal ninein, with a large increase occurring on the DC

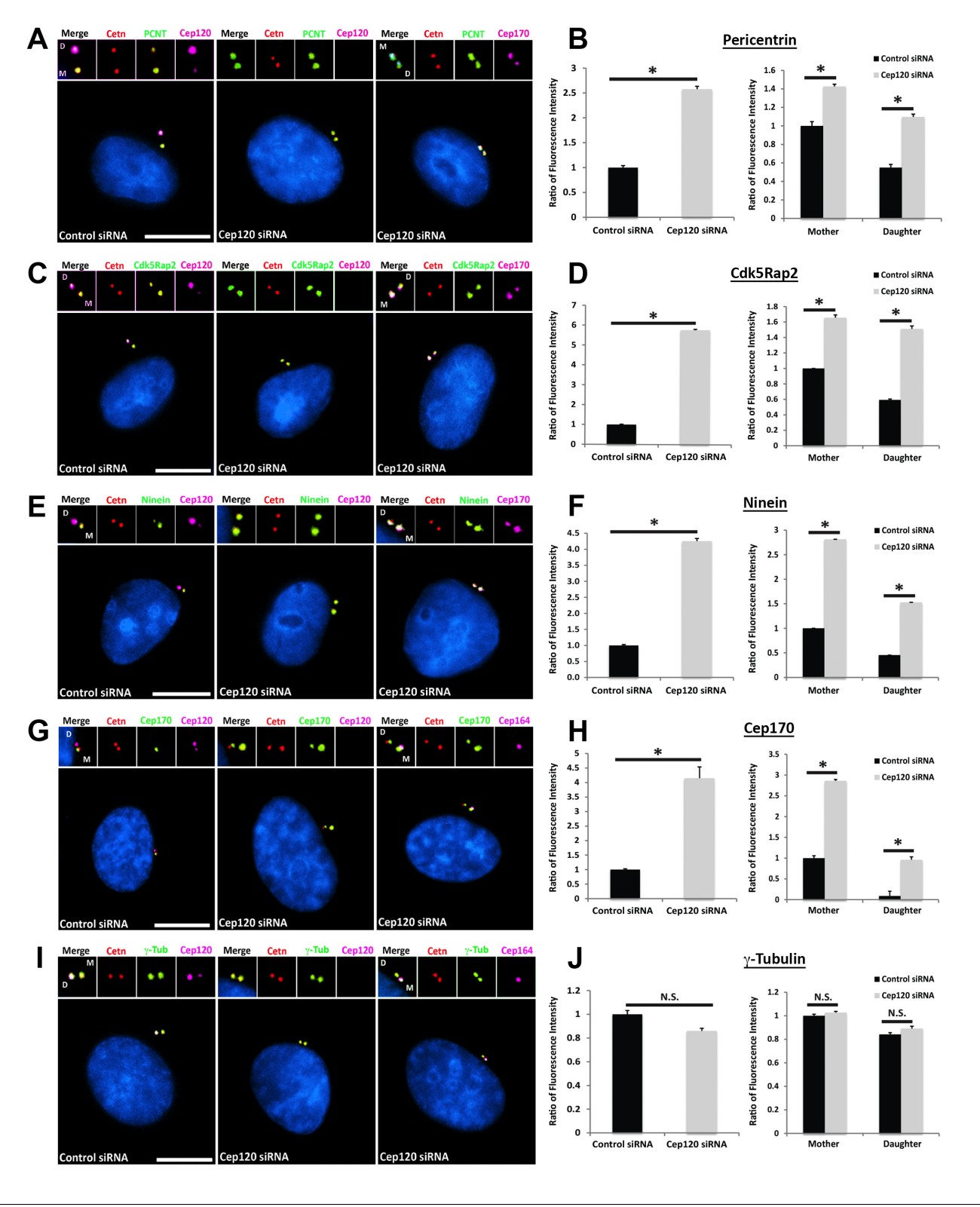

**Figure 2.** Loss of Cep120 in quiescent cells causes accumulation of PCM components on the daughter centriole. MEF cells transfected with control or Cep120-targeting siRNA were immunostained for centrin (centrioles), Cep120, Cep164 or Cep170 (to identify the mother centriole), along with the indicated PCM components. Graphs show quantification of the fluorescence intensity for each PCM protein, normalized to control cells. We noted an overall increase in the abundance of PCM proteins at the centrosome in Cep120-depleted cells, with a large increase observed on the daughter

*Figure 2 continued on next page*

*Figure 2 continued*

centriole. (A–B) Pericentrin: N = 320 (control) and 320 (Cep120) siRNA. (C–D) Cdk5Rap2: N = 400 (control) and 400 (Cep120) siRNA. (E–F) Ninein: N = 430 (control) and 430 (Cep120) siRNA. (G–H) Cep170: N = 480 (control) and 480 (Cep120) siRNA. (I–J) In contrast, there was no significant change in γ-tubulin levels in Cep120-depleted cells. N = 400 (control) and 400 (Cep120) siRNA. Results are averages from three independent experiments; *p<0.05. N.S. = not significant. Scale bars = 10 μm.

DOI: https://doi.org/10.7554/eLife.35439.005

The following figure supplements are available for figure 2:

**Figure supplement 1.** – Rescue of centriole duplication and PCM levels following Cep120 depletion.

DOI: https://doi.org/10.7554/eLife.35439.006

**Figure supplement 2.** – Depletion of Sas6 does not affect PCM levels at the centrosome.

DOI: https://doi.org/10.7554/eLife.35439.007

(*Figure 2E–F*). Importantly, a significant amount of the accumulation occurred around the MC, which indicates that the small pool of Cep120 there also contains some inhibitory function (*Figure 2E–F*). Cep170 is recruited by ninein and shows a similar localization pattern in cells (*Guarguaglini et al., 2005*; *Mazo et al., 2016*), with preferential association with the MC (*Figure 2G*). Loss of Cep120 caused a 4-fold increase in total centrosomal Cep170 levels (*Figure 2G–H*). Similar to ninein, the majority of these gains occurred on the MC, in addition to the significant increase observed on the DC (*Figure 2G–H*). Finally, we measured the relative abundance of γ-tubulin, the core component of the γ-tubulin ring complex involved in microtubule nucleation from the PCM (*Lin et al., 2015*). Intriguingly, we found that loss of Cep120 had no significant effect on the abundance of γ-tubulin at the centrosome, or on mother versus daughter centrioles (*Figure 2I–J*). This suggests that not all components of the PCM may be disrupted upon Cep120 loss. Immunoblot analysis of whole-cell lysates indicated that the increases in centrosomal PCM in Cep120-depleted cells were not due to increased protein expression, but likely changes in localization of the cytosolic pool (*Figure 3*).

To verify the specificity of the siRNA-mediated phenotypes, we generated an expression plasmid expressing an siRNA-resistant version of Cep120, by making six silent mutations in the siRNA-recognition site (GFP-Cep120siRt). Transfection of GFP-Cep120siRt into Cep120-depleted cells rescued

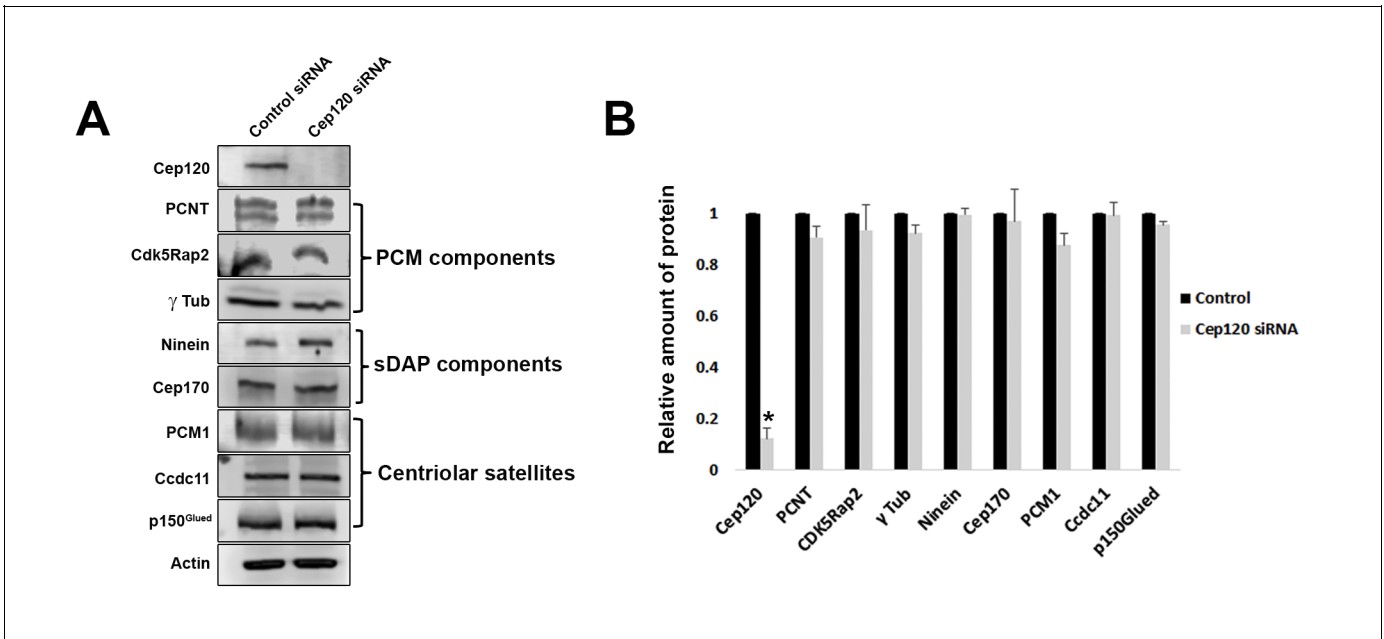

**Figure 3.** Loss of Cep120 does not alter expression levels of centrosomal proteins. (A) Immunoblotting of whole-cell lysates from control and Cep120 siRNA-transfected cells, probed with the indicated antibodies. Actin served as loading control. (B) Quantification of the signal intensity for each protein, normalized to actin. Results are averages of three independent experiments; *p<0.05. Only Cep120 levels show a statistically significant difference.

DOI: https://doi.org/10.7554/eLife.35439.008

the centriole duplication defect in cycling cells (*Figure 2—figure supplement 1A*). Importantly, GFP-Cep120siRt also rescued the increase in pericentrin levels in Cep120-depleted quiescent cells (*Figure 2—figure supplement 1B–C*). We next sought to confirm that the observed changes are specifically due to Cep120 loss in $G_0$, and not an indirect consequence of potentially disrupting centriole duplication. To address this we depleted Sas6, which is typically absent in quiescent cells and is expressed in G1/S-phase where it localizes to, and is critical for, procentriole assembly ([*Dammermann et al., 2004*]; *Figure 2—figure supplement 2A*). Analysis of Sas6-depleted quiescent cells showed no change in the levels of pericentrin, Cdk5Rap2, or Cep170 (*Figure 2—figure supplement 2B–E*). Cep120 and Sas6-depleted cells display loss of centrioles 24 hr after siRNA transfection, resulting in a significant number of cells containing one original parental centriole (*Figure 2—figure supplement 2A*). This allowed us to test whether Cep120 that remains associated with the single daughter centriole in interphase still contained inhibitory activity. We performed this analysis in non-serum starved cells in $G_1$, since this is a stage when Cep120 is still enriched on the daughter centriole; in S-G2-M stages Cep120 enrichment is lost from the daughter centriole (summarized in *Figure 1A*). We compared pericentrin levels in cells treated with Sas6 siRNA (which still maintain Cep120 on the single daughter centriole) to cells transfected with Cep120 siRNA (which have lost Cep120 on the single daughter centriole). We noted a roughly 2-fold increase in pericentrin levels on the single daughter centriole when Cep120 was absent (*Figure 2—figure supplement 2F*). This result is consistent with the hypothesis that the enrichment of Cep120 on the daughter centriole helps to limit PCM recruitment, and that the removal of Cep120 at G1/S is likely mediating accumulation of PCM on that centriole.

Next, we confirmed the inhibitory function of Cep120 by over-expressing exogenous Myc-Cep120, and noted a dose-dependent decrease in centrosomal pericentrin and Cdk5Rap2 (*Figure 4A–B*). Finally, we wanted to identify functionally important domains of Cep120 that are critical for this inhibitory function. Cep120 contains an N-terminal microtubule-binding domain that is essential for directly binding microtubules in vitro (*Lin et al., 2013*; *Sharma et al., 2018*), a CPAP-binding domain essential for centriole duplication (*Comartin et al., 2013*; *Lin et al., 2013*), and a C-terminal coiled-coil domain required for directing localization to centrosomes (*Mahjoub et al., 2010*) (*Figure 4C*). Using a series of deletion constructs that we previously established (*Mahjoub et al., 2010*), we over-expressed truncated versions of the protein and assessed the levels of PCM at the centrosome. Loss of the C-terminal coiled-coil domain failed to inhibit pericentrin levels at the centrosome (*Figure 4C–D*), indicating that localization to centrosomes is important for limiting PCM accumulation. Similarly, deletion of the microtubule-binding domain abrogated Cep120 inhibitory function (*Figure 4C–D*). Collectively, these results indicate that Cep120 limits the levels of PCM proteins at the centrosome (with relatively higher inhibitory activity at the DC), and that both microtubule-binding and coiled-coiled domains are essential for this function.

## Depletion of Cep120 does not alter the localization of distal or subdistal appendages

To test whether loss of Cep120 affected other asymmetrically associated-centriolar proteins, we analyzed the distribution of distal appendage (DAP) and subdistal appendage (sDAP) proteins, which are only found on the MC during quiescence (*Garcia and Reiter, 2016*). Cep164 and Fbf-1 are two core components of the DAP (*Graser et al., 2007*; *Mazo et al., 2016*), and play a critical role in facilitating the attachment of the MC to ciliary vesicles or the plasma membrane, an essential step in cilia formation. There was no difference in Cep164 or Fbf-1 localization between control and Cep120-depleted cells (*Figure 5A–B*). Odf-2 localizes to the sDAP of the MC and is critical for their formation and function (*Ishikawa et al., 2005*; *Mazo et al., 2016*). The asymmetric distribution of Odf-2 was also unchanged upon Cep120 loss (*Figure 5C*). Since ninein and Cep170 are found at the sDAP, in addition to their PCM localization at the proximal region of centrioles (*Mazo et al., 2016*), we sought to determine whether the observed increases in ninein and Cep170 were occurring at the sDAP, PCM or both. 3D-Structured Illumination Microcopy (3D-SIM) of Cep120-depleted cells showed that the increase in ninein and Cep170 on both centrioles was only occurring at the proximal end (*Figure 5D–E*). Together, these results indicate that loss of Cep120 does not affect the asymmetric distribution and localization of DAP and sDAP components, but specifically the PCM.

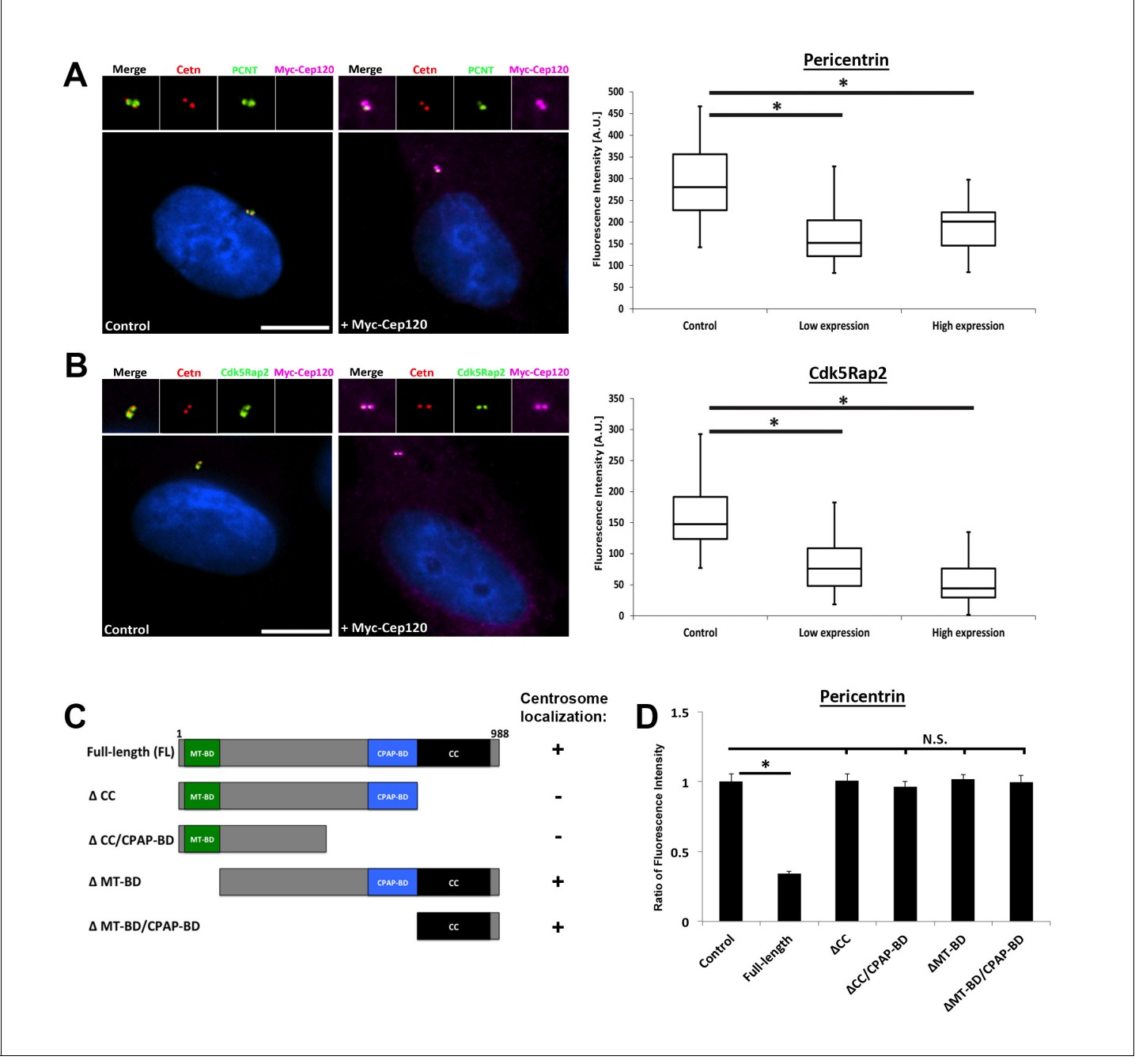

**Figure 4.** Overexpression of exogenous Cep120 results in decreased centrosomal PCM. MEF cells were transfected with plasmid expressing Myc-Cep120, serum-starved for 24 hr, fixed and stained for Myc, centrin (centrioles), and the PCM components (**A**) Pericentrin or (**B**) Cdk5Rap2. Graphs show quantification of the fluorescence intensity for each PCM protein at the centrosome, in control cells (untransfected), and cells expressing low (1–2 fold compared to endogenous levels) versus high (2–3 fold) levels of centrosomal Myc-Cep120 (based on fluorescence intensity). We noted a dose-dependent decrease in pericentrin and Cdk5Rap2 levels correlating with increased exogenous Cep120 expression. Pericentrin: N = 180 (control), and 120 (Myc-Cep120). Cdk5Rap2: N = 165 (control), and 130 (Myc-Cep120). Results are averages of three independent experiments; *p<0.05. Scale bars = 10 µm. (**C**) Schematic of Cep120-GFP deletion constructs. CC = coiled coil; MT-BD = microtubule binding domain; CPAP-BD = CPAP binding domain. Domains required for centrosome localization were previously described (*Mahjoub et al., 2010*). (**D**) MEF cells were transfected with plasmids expressing Cep120-GFP deletion constructs, serum-starved for 24 hr, fixed and stained for GFP, centrin, and pericentrin. Graphs show quantification of the relative fluorescence intensity for pericentrin at the centrosome. Only over-expression of the full-length protein reduced centrosomal pericentrin levels. Results are averages of two independent experiments; *p<0.05.

DOI: https://doi.org/10.7554/eLife.35439.009

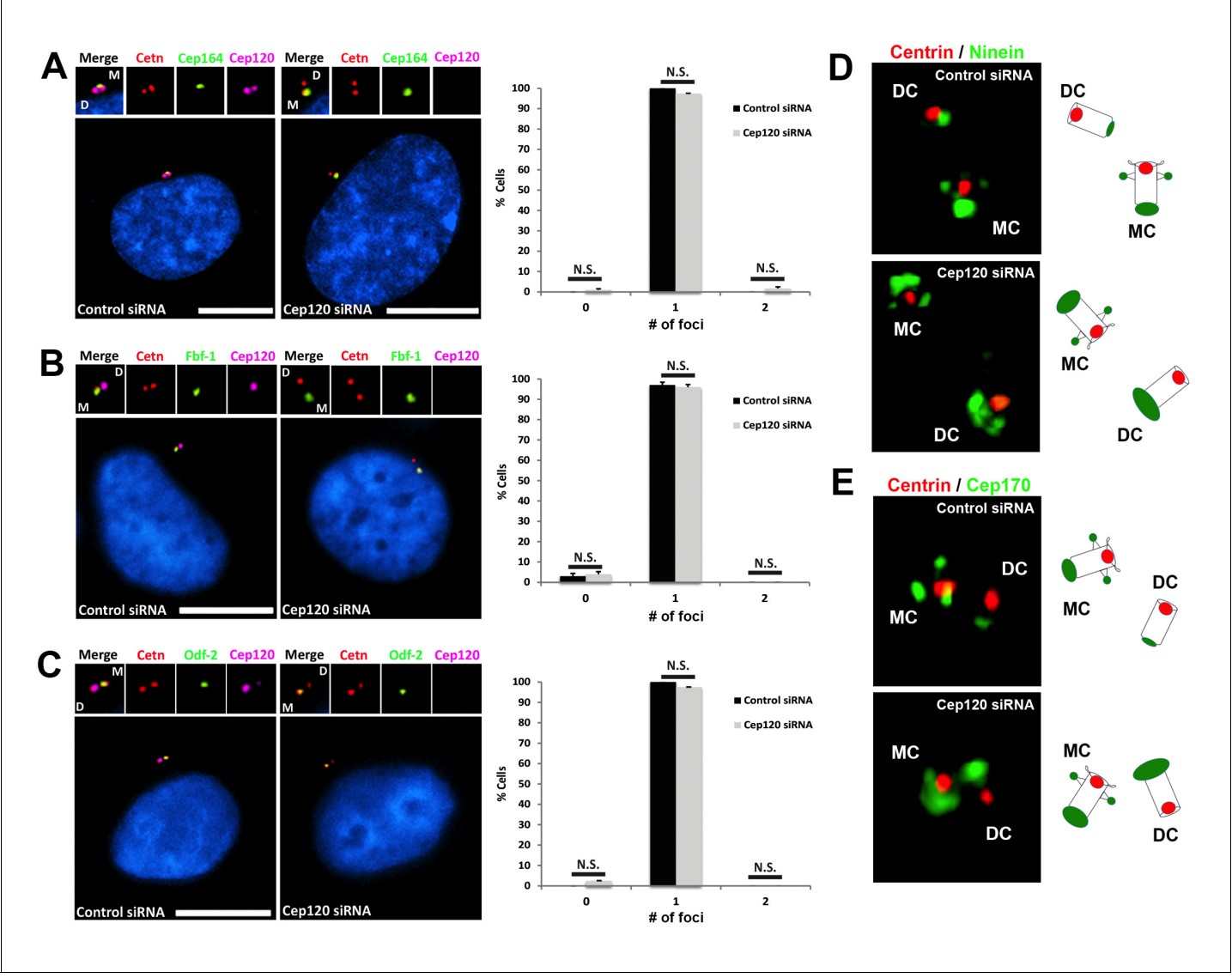

**Figure 5.** Depletion of Cep120 does not affect the localization of distal or subdistal appendage proteins. MEF cells were transfected with control or Cep120-targeting siRNA, and immunostained for centrin (centrioles), Cep120, the distal appendage proteins (**A**) Cep164 (N = 300 (control) and 300 (Cep120) siRNA) and (**B**) Fbf-1 (N = 300 (control) and 300 (Cep120) siRNA), or (**C**) the subdistal appendage protein Odf-2 (N = 200 (control) and 200 (Cep120) siRNA). Quantification of the number of foci per cell indicates that Cep120-depleted cells mainly contain one focus of each protein, which is similar to control siRNA-treated cells. Results are averages of three independent experiments; *p<0.05. N.S. = not significant. Scale bars = 10 µm. (**D–E**) 3D-SIM images of centrioles from control and Cep120-depleted MEF cells, immunostained for centrin (centrioles), ninein or Cep170. Schematics (right) represent relative positions of each marker on mother versus daughter centrioles.

DOI: https://doi.org/10.7554/eLife.35439.010

## Increased microtubule nucleation and dispersal of centriolar satellites upon Cep120 loss

To determine whether the increased PCM in Cep120-depleted cells affects centrosome function during quiescence, we analyzed microtubule assembly post-depolymerization. Microtubules were disassembled by a combination of nocodazole and cold treatment, and nucleation assessed at different time points during re-assembly by immunostaining for α-tubulin. There were higher levels of microtubule re-growth in Cep120-depleted quiescent cells, as evidenced by larger microtubule asters (*Figure 6A–B*). Moreover, staining for the plus-end tracking protein EB1 showed more abundant EB1-positive comets focused around the centrosome (*Figure 6C*), indicating a higher number of

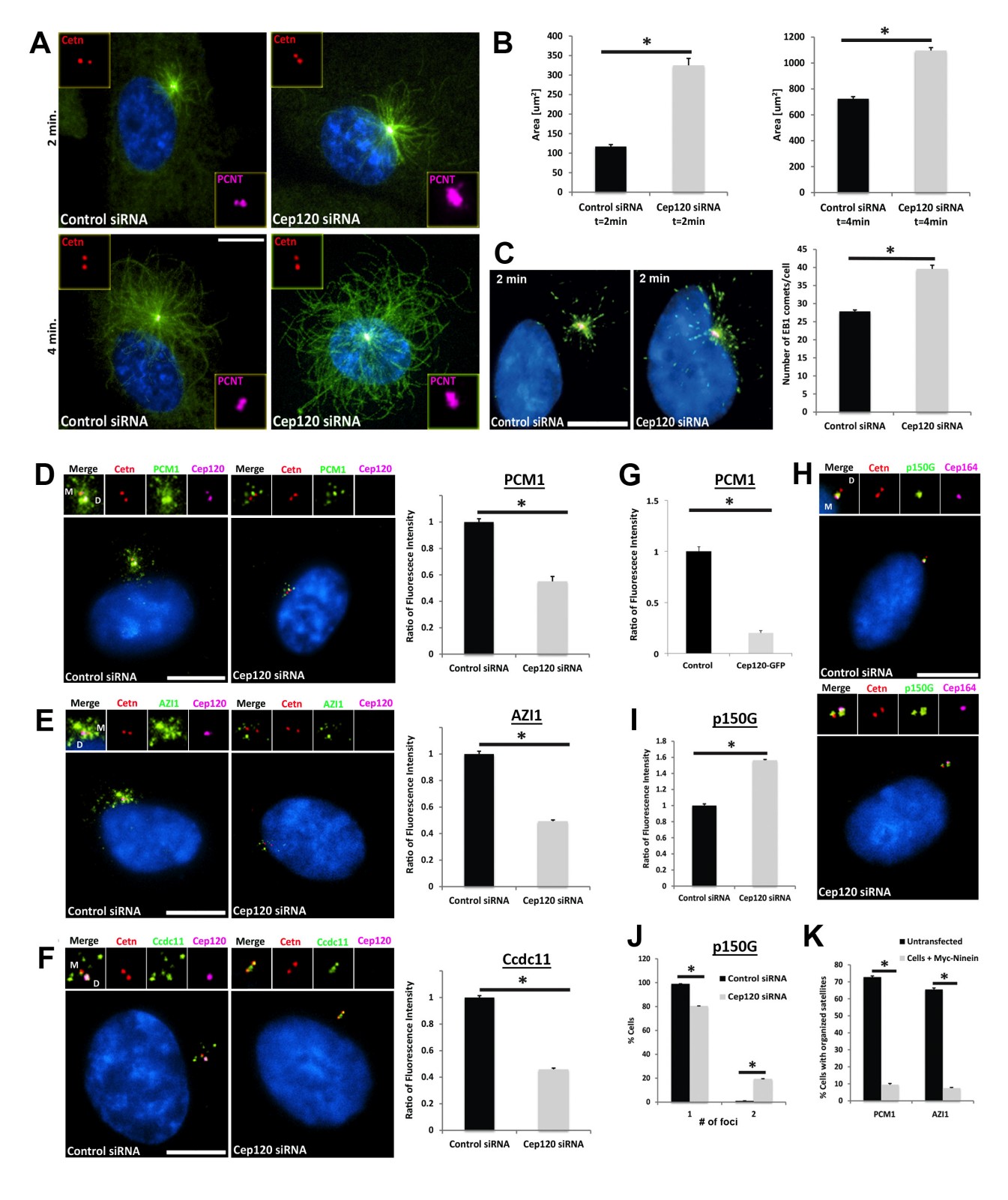

**Figure 6.** Increased microtubule nucleation and dispersal of centriolar satellites in Cep120-depleted cells. (**A**) MEFs transfected with control or Cep120-siRNA for 48 hr were incubated with 0.1 μg/mL nocodazole on ice for 1 hr, washed and incubated with warm (37°C) media for the indicated times, fixed and stained for centrin (centrioles), pericentrin (PCM), α-tubulin (microtubules), and DAPI (DNA). (**B**) Quantification of microtubule aster size during regrowth. N = 400 (control) and 400 (Cep120) siRNA. (**C**) Cells at 2 min of regrowth were fixed and stained for α-tubulin (red), EB1 (green) and DAPI

*Figure 6 continued on next page*

Figure 6 continued

(blue). Graph denotes average number of EB1-positive foci at each centrosome. N = 300 (control) and 240 (Cep120) siRNA. (D–F) Loss of Cep120 causes mislocalization (dispersal) of centriolar satellites. Graphs show quantification of the fluorescence intensity for each satellite protein, normalized to control cells. (D) PCM-1: N = 375 (control) and 275 (Cep120) siRNA. (E) Azi-1: N = 215 (control) and 175 (Cep120) siRNA. (F) Ccdc11: N = 130 (control) and 134 (Cep120) siRNA. (G) Overexpression of exogenous Cep120 disrupts centriolar satellite localization. MEF cells were transfected with plasmid expressing Cep120-GFP, serum-starved for 24 hr, fixed and stained for GFP, centrin (centrioles) and PCM-1. Graph shows the fraction of cells with organized satellite protein localization at the centrosome. N = 200 cells per sample. Results are averages of two independent experiments; *p<0.05. (H–J) Loss of Cep120 results in accumulation of p150$^{Glued}$ at the centrosome. N = 315 (control) and 285 (Cep120) siRNA. Results are averages from three independent experiments; *p<0.05. Scale bars = 10 µm. (K) Overexpression of exogenous ninein disrupts centriolar satellite localization. MEF cells were transfected with plasmid expressing Myc-ninein, serum-starved for 24 hr, fixed and stained for Myc, centrin (centrioles), and the satellite proteins PCM-1 and Azi-1. Graph shows the fraction of cells with organized satellite protein localization at the centrosome. PCM-1: N = 300 (control untransfected) and 100 (Myc-ninein). Azi-1: N = 300 (control untransfected) and 100 (Myc-ninein). Results are averages of three independent experiments; *p<0.05. Scale bars = 10 µm.

DOI: https://doi.org/10.7554/eLife.35439.011

nucleated microtubules from the centrosome. Next, we tested whether microtubule-dependent centrosomal trafficking was affected. Centriolar satellites (CS) are non-membranous complexes found in the vicinity of the centrosome in mammalian cells (*Hori and Toda, 2017*). They play pivotal roles in centrosome assembly, primary cilium formation and function, via the delivery of components from the cytoplasm to the centrosome. These trafficking functions of CS are dependent on an intact, organized microtubule array (*Hori and Toda, 2017*). Remarkably, we discovered that multiple components of CS were dispersed in Cep120-depleted cells, including the core scaffolding protein PCM-1, as well as regulators of ciliogenesis Azi-1 and Ccdc11 ([*Kubo et al., 1999*; *Silva et al., 2016*; *Staples et al., 2012*]; *Figure 6D–F*). Similarly, overexpression of exogenous Cep120-GFP, which leads to a decrease in pericentrin levels (*Figure 4D*) also resulted in dispersal of PCM-1 (*Figure 6G*), indicating that PCM homeostasis is key for satellite localization and distribution.

To understand why these CS were mislocalized, we immunostained cells for a component of the cytoplasmic dynein-dynactin motor complex p150$^{Glued}$, which regulates CS localization and centrosome-directed movement (*Kim et al., 2004*). In control cells, p150$^{Glued}$ was predominantly localized to the MC (*Figure 6H*), as previously shown (*Kodani et al., 2010*). We observed a ~ 1.6 fold increase in centrosomal p150$^{Glued}$ in Cep120-depleted cells, with the protein accumulating on the DC, which is rarely seen in control cells (*Figure 6H–J*). We reasoned that the satellite dispersal phenotype might be caused by the increased ninein levels, which recruits p150$^{Glued}$ to the centrosome (*Mazo et al., 2016*). Indeed, overexpression of exogenous Myc-ninein caused dispersal of PCM-1 and Azi-1 (*Figure 6K*). Collectively, these results suggest that loss of Cep120 disrupts the localization of CS, due to changes in PCM abundance, microtubule organization, and dynein-dependent trafficking of centrosomal proteins.

## Loss of Cep120 disrupts ciliary assembly and signaling during quiescence

Since satellites play essential roles in ciliogenesis, we hypothesized that dispersal of CS upon Cep120 loss would disrupt the formation and/or function of cilia during quiescence. Indeed, Cep120-depleted cells exhibit a > 60% decrease in cilia formation compared with controls (*Figure 7A–B*). Moreover, the fraction of cells that did assemble/retain cilia formed ones that were aberrant in length (*Figure 7C–D*), another well-established indicator of defective ciliary protein trafficking. Once again, these defects were specific to loss of Cep120, as control Sas6-depleted cells displayed normal CS localization (*Figure 7—figure supplement 1A*) and were competent to form cilia (*Figure 7—figure supplement 1B*). Next, we assessed whether ciliary signaling was affected in the fraction of Cep120-depleted cells that still formed cilia. We focused on the transmembrane protein smoothened (Smo), which translocates into the cilium in response to sonic hedgehog (Shh) ligand (*Corbit et al., 2005*). The amount of Smo per unit length cilium was significantly reduced in Cep120-depleted cells (*Figure 7C–E*). This suggests that in addition to assembly, the signaling functions of cilia may be attenuated upon Cep120 loss. The defects in ciliogenesis were not due to mislocalization of intraflagellar transport (*Jurczyk et al., 2004*) machinery (*Figure 7—figure supplement 1C–*

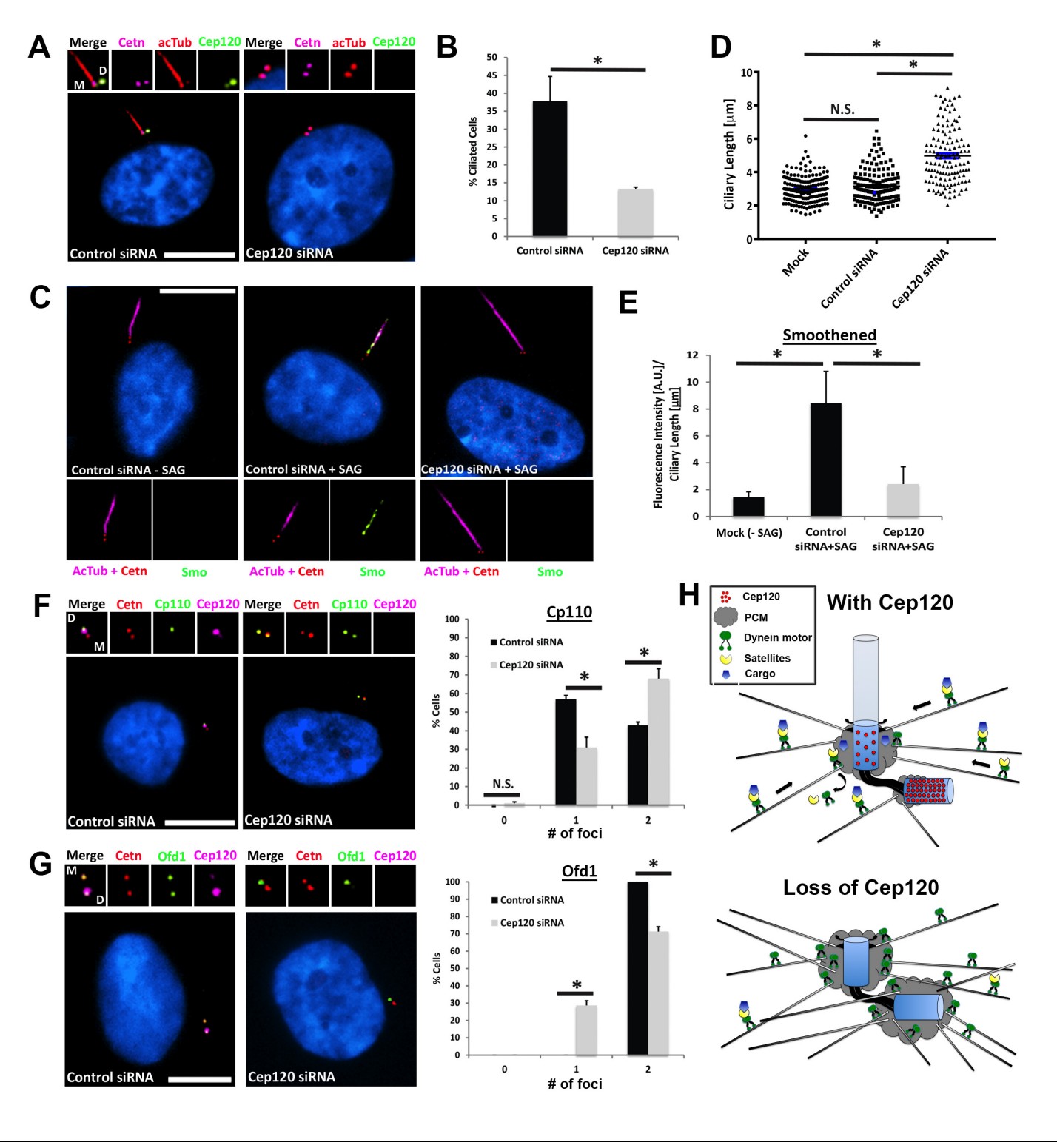

**Figure 7.** Loss of Cep120 disrupts ciliary assembly and signaling. (**A**) Quiescent MEF cells stained with antibodies against Cep120, centrin (centrioles), and acetylated α-tubulin (cilia). (**B**) Quantification of control (N = 300) and Cep120-depleted (N = 300) cells that form cilia. (**C**) Endogenous Smo localization in control and Cep120-depleted cells treated with the Shh pathway agonist SAG. (**D**) Distribution of ciliary length in untransfected (N = 185), control siRNA (N = 160), and Cep120-depleted (N = 132) cells. (**E**) Ratio of ciliary Smo intensity per unit length cilium in control (N = 156) and Cep120-depleted (N = 136) cells treated with SAG. (**F**) CP110 predominantly marks the daughter centriole in control quiescent cells. In Cep120-depleted cells, there is higher incidence of CP110 association with both centrioles, correlating with the decrease in ciliogenesis. N = 300 (control) and 300 (Cep120)

*Figure 7 continued on next page*

*Figure 7 continued*

siRNA. (**G**) Ofd1 localizes to both centrioles in control cells, which is disrupted in Cep120-depleted cells. N = 300 (control) and 300 (Cep120) siRNA. Results are averages from three independent experiments; *p<0.05. N.S. = not significant. Scale bars = 10 µm. (**H**) Model depicting PCM accumulation, increased microtubule nucleation, aberrant centriolar satellite organization and ciliogenesis defects upon Cep120 loss in quiescent cells.

DOI: https://doi.org/10.7554/eLife.35439.012

The following figure supplement is available for figure 7:

**Figure supplement 1.** – Depletion of Sas6 does not affect centriolar satellite organization or ciliogenesis.

DOI: https://doi.org/10.7554/eLife.35439.013

*D*), nor the transition zone protein Cep290 (*Figure 7—figure supplement 1E–F*), which is at CS then localizes to the transition zone upon ciliogenesis (*Kim et al., 2008*).

Finally, we examined the localization of essential ciliogenesis factors that are dependent on CS for their centrosomal localization. CP110 is a centriolar distal end capping protein that interacts with CS and antagonizes ciliogenesis; its removal from the MC is a critical step in ciliogenesis (*Spektor et al., 2007*). Quiescent cells that form cilia contain CP110 only on the DC, as it is removed from the MC (*Figure 7F*). Intriguingly, we noted a significant increase in the fraction of cells that still contained CP110 on both the MC and DC (*Figure 7F*). Finally, we examined the localization of the ciliopathy protein Ofd1, which is essential for ciliogenesis, interacts with PCM-1 at satellites, and localizes to the distal end of both centrioles ([*Lopes et al., 2011*]; *Figure 7G*). Cep120-depleted cells showed a decrease in the number of Ofd1 foci at centrioles, indicating a defect in centriolar targeting of the protein (*Figure 7G*). Together, these results show that loss of Cep120 during quiescence is detrimental to ciliary assembly and function, likely due to aberrant CS-dependent trafficking of proteins to the centrosome.

## Discussion

In this study, we have demonstrated that Cep120 plays an important inhibitory role at the centrosome, by regulating PCM levels during quiescence. Loss of Cep120 initiates a series of changes in protein localization and abundance that ultimately result in defective ciliary assembly and function (*Figure 7H*). The first step involves changes in the maturation state of the daughter centriole, on which Cep120 is preferentially enriched. Centriole maturation is a multi-step process that includes accumulation of PCM material, assembly of appendage structures, recruitment of proteins essential for ciliary assembly, and elimination of inhibitory/licensing factors (such as CP110) of ciliogenesis. Although most of these components are asymmetrically localized on mother versus daughter centrioles, our results show that Cep120 mainly controls the PCM stage of the maturation process. Removal of Cep120 did not result in complete maturation of the daughter centriole, nor enhance its ability to build a primary cilium. This suggests that the centriole maturation pathway regulated by Cep120 during quiescence is different than the one controlled by Plk1, a kinase that triggers centrosome maturation as cells enter mitosis (*Mahen and Venkitaraman, 2012*). Indeed, over-expression of Plk1 in $G_1$ can fully induce daughter centriole maturation independently of Cep120 loss, resulting in the assembly of appendages and the formation of a secondary primary cilium (*Kong et al., 2014*). An alternative possibility is that Cep120 may be one of many targets of Plk1 and thus its depletion would not be expected to phenocopy Plk1 overexpression. However, inducing expression of Plk1 in $G_1$ did not cause removal of Cep120 from the maturing daughter centriole (*Kong et al., 2014*), suggesting that cells may have evolved an alternative pathway by which they can regulate centriole maturation during quiescence, controlled by protein(s) that localize to the daughter centriole.

Depletion of Cep120 resulted in preferential accumulation of PCM components on the daughter centriole. However some PCM proteins, such as ninein and Cep170, showed higher levels of accumulation on the mother centriole. This indicates that the small pool of Cep120 that remains associated with the mother centriole also contains some inhibitory function, and suggests that the levels of certain PCM proteins on the mother centriole are also 'fine-tuned' during quiescence. One intriguing reason for this type of control may be to ensure that the mother centriole (and the centrosome in general) assembles a microtubule network that specifically helps target ciliary proteins to the mother centriole (*Figure 7H*). Indeed, we found that the elevated PCM negatively impacts centrosome homeostasis by disrupting the nucleation and organization of the microtubule cytoskeleton,

potentially causing a 'traffic jam' that interrupts dynein-dependent movement of CS complexes. It is well established that the organization, localization and trafficking functions of CS are particularly sensitive to defects in the microtubule cytoskeleton (*Hori and Toda, 2017*). These changes in turn disrupted ciliary assembly and signaling, a process that requires CS-dependent delivery of essential factors to the centrosome.

How does Cep120 regulate the asymmetric distribution of PCM proteins at the centrosome? Using immuno-electron microscopy, we previously showed that Cep120 resides on the outer wall of the centriolar microtubules (*Mahjoub et al., 2010*). Subsequent studies using superresolution microscopy also positioned Cep120 near the centriolar microtubules, placing it in a layer proximal to the centriole wall and surrounded by the PCM proteins (*Lawo et al., 2012*; *Mennella et al., 2014*). Moreover, Cep120 contains an N-terminal microtubule-binding domain that was shown to be essential for directly binding microtubules in vitro (*Lin et al., 2013*). Intriguingly, mutations in this microtubule-binding domain are pathogenic, and cause ciliopathies such as Jeune and Joubert syndrome (*Roosing et al., 2016*; *Shaheen et al., 2015*). Thus, one interesting possibility is that Cep120 binds to and occupies sites on the surface of the centriolar microtubule wall, which could inhibit the association of PCM proteins in those locations. The loss of Cep120 from the daughter centriole at G1/S could potentially 'free' these sites and permit the accumulation of PCM proteins. The mechanism(s) that establish the asymmetric localization of Cep120 on mother and daughter centrioles remain unknown, and will be the focus of our future studies.

In sum, our results highlight a new function for daughter centriole-associated proteins that is independent of their role in centriole duplication and assembly. Furthermore, these data illuminate the potential role(s) of other daughter centriole-enriched proteins, such as centrobin and Neurl4 (*Li et al., 2012*; *Zou et al., 2005*), in quiescent cells. Finally, our findings establish the important and exquisitely sensitive control of centrosome PCM homeostasis and its relationship to primary cilium assembly and signaling, and uncover a potentially new type of molecular defect that may underlie the pathogenesis of ciliopathies such as Jeune and Joubert syndrome.

## Materials and methods

### Cell culture and media

Mammalian cells used in this study were mouse embryonic fibroblasts (MEF), hTERT-immortalized human retinal pigment epithelial cells (RPE-1), NIH3T3 fibroblasts and NIH3T3 cells stably expressing Cep120-GFP (*Mahjoub et al., 2010*). The cells were originally obtained from the ATCC and their identity authenticated. We have tested for mycoplasma contamination using commercially available kits. Cells were grown in DMEM medium (Corning, cat. # 10–013-CV) supplemented with 10% FBS (Atlanta Biologicals, cat. # S11150) and 1% penicillin-streptomycin solution (Gibco, cat. # 15140–122). For serum-starvation experiments, cells were incubated for 24–48 hr in DMEM supplemented with 0.5% FBS.

### Plasmids, siRNA and transfections

Cep120 constructs used in this study have been previously described (*Mahjoub et al., 2010*): pCS2-Myc-Cep120 (full-length, aa 1–988), pCS2-Cep120-GFP (full-length; FL), pCS2-Cep120-GFP ΔCC (aa 1–700), pCS2-Cep120-GFP ΔCC/CPAP-BD (aa 1–450), pCS2-Cep120-GFP ΔMT-BD (aa 101–988), and pCS2-Cep120-GFP ΔMT-BD/CPAP-BD (aa 700–988). For experiments requiring overexpression of ninein, a Myc-ninein plasmid that was previously generated was used (*Firat-Karalar et al., 2014a*). MEF cells were transfected with 1–5 µg of the appropriate plasmid DNA using 10 µL of Lipofectamine 2000 (Life Technologies, cat. # 11668–027) according to the manufacturer's protocol. Medium was changed 24 hr after transfection, and quiescence induced by incubating cells in low FBS (0.5%) for another 24 hr. For experiments using siRNA-mediated depletion of Cep120, a previously characterized (*Comartin et al., 2013*) synthetic oligonucleotide targeting the C-terminal sequence of the human and mouse Cep120 gene was obtained from GE Healthcare Dharmacon Inc. (siGENOME collection, cat. # D-016493–01). The siRNA sequence is: 5'- GAUGAGAACGGGUGUG UAU-3'. For depletion of Sas6, the previously characterized synthetic oligonucleotide siRNA#3 (*Leidel et al., 2005*) targeting the C-terminal domain of human Sas6 was obtained (Dharmacon, siGENOME collection, cat. # D-019156–03). The siRNA sequence is: 5'-GCACGUUAAUCAGC

UACAA-3'. A Silencer Select Negative Control siRNA (Life Technologies, cat. # 4390844) was used as control. MEF or RPE-1 cells were seeded at 50–80% confluence and transfected with 100–200 nM of siRNA oligonucleotide using Lipofectamine RNAiMAX (Life Technologies, cat. # 13778–075) according to manufacturer's protocol. Depletion of Cep120 or Sas6 protein was observed 48 hr post-transfection by immunofluorescence and by immunoblotting. Ciliogenesis was induced at 24 hr post-depletion by shifting cells from high (10%) to low (0.5%) serum-containing medium for another 24 hr. For rescue experiments, we generated a siRNA-resistant GFP-Cep120 expression plasmid (pLVX-Puro-GFP-Cep120siRt) that contains six silent mutations introduced in the siRNA-targeting sequence. The changes to the siRNA-targeting sequence are (altered sites marked in bold): 5'-**A**AT-GAG**G**ACA**G**GA**G**T**A**TA**C**-3'. MEF cells were transfected with 5 µg of pLVX-Puro-GFP-Cep120siRt plasmid using 10 µL of Lipofectamine 2000 24 hr after siRNA transfection, and quiescence was induced by incubating cells in low-serum medium FBS for a further 24 hr.

## Cep120 antibody generation and immunofluorescence microscopy

To generate a rat polyclonal Cep120 antibody, a C-terminal fragment of the mouse Cep120 gene (amino acids 860–988) was PCR amplified and subcloned into pET-21a (+) vector (Novagen, cat. # 69740–3) at unique NdeI and XhoI sites. Soluble recombinant Cep120-His6 protein was expressed in BL21-CodonPlus(De3)-RIL competent cells (Agilent Technologies, cat. # 230280) and purified on His-PurTM Cobalt Resin (Pierce, cat. # 89964) following manufacturer's protocols. 8 mg of antigen was used to inject three rats (PrimmBiotech, Inc.). Polyclonal anti-Cep120 antibody from two of the animals showed specificity for Cep120 by immunoblot analysis of whole cell lysates (used at a dilution of 1:5,000-1:10,000; *Figure 1—figure supplement 1*) and by immunofluorescence microcopy (used at 1:2,000; *Figures 1–7*). For indirect immunofluorescence, cells were grown on glass coverslips (Electron Microscopy Sciences, cat. # 72230–01) coated with 1 mg/mL poly-L-Lysine (Sigma, cat. # P1274), fixed in either ice-cold methanol at −20°C for 10 min or with 4% paraformaldehyde (Electron Microscopy Sciences, cat. # 15710) in PBS at room temperature for 10 min. Cells were rinsed twice with PBS, permeabilized with 0.1% Triton X-100 in PBS (PBS-T), and blocked for 1 hr at room temperature with 3% BSA (Sigma, cat. # A2153) in PBS-T. Samples were incubated with primary antibodies for 1–2 hr at room temperature (see *Supplementary file 1* for complete list of antibodies used). Alexa Fluor dye–conjugated secondary antibodies (Life Sciences) were used at a dilution of 1:500 at room temperature for 1 hr. Nuclei were stained with DAPI, and mounted in Mowiol (Sigma, cat. # 81381) containing N-propyl gallate (Sigma, cat. # 02370). Images were captured using a Nikon Eclipse Ti-E inverted confocal microscope equipped with a 60X (1.4 NA) or 100X (1.45 NA) Plan Fluor oil immersion objective lens (Nikon, Melville, NY). Three-dimensional super-resolution microscopy (3D-SIM) images were captured on an inverted Nikon Ti-E microscope using 100X oil objective. Optical z-sections were separated by 0.216 µm, and 3D-SIM images were processed and reconstructed using the Nikon Elements AR 4 Software.

## Analysis of centrosomal and ciliary proteins

Quantification of fluorescence intensity of the various centrosomal markers (Pericentrin, Cdk5Rap2, ninein, γ-tubulin, Cep170) was performed by measuring the total fluorescent intensity of a given marker within a 2–5 µm circular region of interest (ROI) centered on the centrosome, using Nikon Elements AR 4.20 Software. Background fluorescence values were measured using an ROI of identical size in the near proximity of each centrosome and subtracted from the centrosomal measurements. For comparison of signal intensity at mother versus daughter centrioles, two separate ROIs of 0.5–1.0 µm were drawn around each centriole. Background fluorescence subtraction was performed as described above. In both cases, the fluorescence signal of centrin was simultaneously measured and used as an internal control, and the fluorescent values of each marker normalized to the unchanged centrin signal. Centriolar satellite localization and dispersal was determined by measuring the fluorescence intensity of PCM-1, Azi-1, Ccdc11 and p150Glued within a 2–16 µm circular region of interest (depending on the protein) as described. The relative localization of various centriolar/centrosomal proteins on mother versus daughter centrioles was determined by comparing control and Cep120-depleted cells for each marker individually. These include Cep164, Fbf-1 (distal appendages), Cep170, Odf-2, (subdistal appendages), Cep290 and Ofd1 (transition zone), Cp110 (centriole cap), and IFT88 (intraflagellar transport protein). For overexpression of exogenous

Cep120, we compared the ratio of exogenous and endogenous protein at the centrosome by immunostaining cells with anti-Myc and anti-Cep120 antibodies. We consider 1-2X the relative amount of exogenous Cep120 (compared to endogenous protein) 'low' overexpression, while 'high' overexpression ranged from 2-3X.

Primary cilium assembly was determined by counting the percentage of control and Cep120-depleted cells that formed a cilium following serum-starvation. To stimulate sonic hedgehog (Shh) signaling, control and Cep120-depleted MEFs were treated with SAG (500 nM, Enzo Life Sciences, cat # ALX-270–426 M001) in low-serum medium for 28 hr. Cells were fixed in 2% PFA for 10 min at room temperature, washed and post-fixed in ice-cold methanol for 10 min. Samples were incubated with anti-Smoothened overnight at 4°C. Cells were washed and stained the next day for centrin (to mark centrioles) and acetylated α-tubulin (to mark primary cilia). Ciliary length was determined using acetylated α-tubulin as marker, and smoothened fluorescence intensity was measured by selecting a region encompassing the length of each cilium. The ratio of smoothened per unit length cilium was determined using Excel (Microsoft Office), and graphs generated with GraphPad Prism 7.02 Software (GraphPad Software Inc., La Jolla, CA, USA). More than 130 cilia were scored per experiment, and averages determined from two independent experiments.

For analysis of microtubule density during regrowth, serum-starved MEFs were treated with 0.1 µg/mL nocodazole (SIGMA, cat # M1404) for 1 hr on ice to depolymerize microtubules. Coverslips were washed three times, incubated with 37°C low-serum growth medium to allow microtubule regrowth, and then fixed at 0, 2 and 4 min intervals. Cells were stained with anti-α-tubulin to visualize microtubule asters, centrin to visualize centrioles, and EB-1 to count microtubule +tips. The average diameter/radius of growing microtubule asters was determined by drawing a circular ROI centered on the centrosome, and the distal end of the longest microtubule tip in each cell. In parallel, the number of EB-1-positive foci emanating from the growing microtubule aster was determined. More than 150 cells were scored per experiment, and the average values quantified from three independent experiments.

## Cell extracts and western blotting

To obtain whole-cell extracts for immunoblotting, quiescent cells were washed in PBS and lysed by the addition of SDS sample buffer. Proteins were resolved on 6–8% SDS-PAGE gels, depending on protein size. Immunoblots were incubated with primary antibodies overnight at 4°C (see Supplementary File 1). Species-specific horseradish peroxidase-conjugated secondary antibodies were: mouse anti-Rabbit-HRP (1:10,000; Sigma, cat # R3155), donkey anti-Mouse-HRP (1:10,000, Jackson Immuno Research, cat # 715-035-150), donkey anti-Guinea pig-HRP (1:10,000; Jackson Immuno Research, cat # 706-035-148) and AffiniPure donkey anti-Rat-HRP (1:10,000; Jackson Immuno Research, cat # 712-035-153). Blots were imaged on a C-DiGit Blot Scanner (LI-COR, cat # 3600–000), and signal intensities for each protein quantified and normalized to actin.

## Cell cycle analysis

Two independent assays were utilized to establish the cell cycle state of control and Cep120-depleted cells: (1) quantitation of DNA content was performed by flow cytometry. MEF cells were transfected with Cep120 or control siRNA, and cycling cells harvested after 24 hr. Alternatively, siRNA-transfected cells were incubated in low-serum medium for 48 hr (quiescence stage). Cells were pelleted, resuspended in 0.5 ml DPBS, and permeabilized with 70% ice-cold ethanol for 2 hr. Samples were washed in cold PBS and resuspended in 0.5 mL of 20 µg/ml Propidium Iodide (Life Technologies, cat # P3566) supplemented with 0.1% Triton X-100 and 2 µg/ml RNase A (Sigma, cat # R4875), and incubated for 30 min. at RT. Before FACS analysis, cell aggregates were removed by passaging through a filter (CellTrics, 50 µm strainer, cat # 04-004-2327). At least 10,000 cells/sample were analyzed on a digitally modified DxP FACScan (BD Biosciences, San Jose, CA and Cytek Development, Fremont, CA) and the data analyzed using FlowJo software version 10.2 (FlowJo, LLC, Ashland, OR); (2) Control and Cep120-depleted MEFs were serum-starved for 48 hr to induce quiescence, fixed and stained for the proliferation marker Ki-67. The fraction of Ki-67-positive cells was determined by counting more than 100 cells per experiment, and average values determined from three independent experiments.

## Statistical analyses

Data from fluorescence intensity measurements, number of centriolar foci, centriolar satellite localization and ciliogenesis assays were examined by unpaired two-tailed Student's *t*-test (p<0.05). Data from Shh assays were examined by one-way analysis of variance (ANOVA) for significance (p<0.05). When significant differences were detected, treatment means were further analyzed by Tukey's honest significant difference. All graphs were created using either Excel or GraphPad Prism 7.02 software. Data are reported as mean ± SEM.

# Acknowledgements

We thank W.C. Eades (Siteman Cancer Center Flow Cytometry Core, Washington University, Grant #P30 CA91842) for help with FACS analysis. We also thank T. Stearns (Stanford University) for generously sharing the Myc-ninein plasmid, anti-PCM-1 and anti-Cep164 antibodies, J. Sillibourne and M Bornens (Insitut Curie, Paris, France) for the anti-Ninein antibody, and B. Franco (Telethon Institute of Genetics and Medicine, Italy) for the Ofd1 antibody. Some image acquisition and data analyses were performed in part through the use of Washington University Center for Cellular Imaging (WUCCI) supported by Washington University School of Medicine, The Children's Discovery Institute of Washington University and St. Louis Children's Hospital (CDI-CORE-2015–505) and the National Institute for Neurological Disorders and Stroke (NS086741). This project was supported by funding from the National Institute of Diabetes and Digestive and Kidney Diseases (R01-DK108005) to M.R. M.

# Additional information

### Funding

| Funder | Grant reference number | Author |
|---|---|---|
| National Institute of Diabetes and Digestive and Kidney Diseases | R01-DK108005 | Moe R Mahjoub |

The funders had no role in study design, data collection and interpretation, or the decision to submit the work for publication.

### Author contributions

Ewelina Betleja, Conceptualization, Formal analysis, Validation, Investigation, Visualization, Methodology, Writing—original draft, Writing—review and editing; Rashmi Nanjundappa, Formal analysis, Investigation; Tao Cheng, Investigation, Methodology, Generated the siRNA-resistant Cep120-GFP plasmid, Helped with the rescue experiments; Moe R Mahjoub, Conceptualization, Formal analysis, Supervision, Funding acquisition, Investigation, Methodology, Writing—original draft, Writing—review and editing

### Author ORCIDs

Rashmi Nanjundappa (iD) http://orcid.org/0000-0003-3621-4628
Moe R Mahjoub (iD) http://orcid.org/0000-0001-8129-7464

### Decision letter and Author response

Decision letter https://doi.org/10.7554/eLife.35439.017
Author response https://doi.org/10.7554/eLife.35439.018

# Additional files

### Supplementary files

• Supplementary file 1. List of antibodies used in this study, including fixation and concentrations used for immunofluorescence staining and immunoblotting.
DOI: https://doi.org/10.7554/eLife.35439.014

• Transparent reporting form
DOI: https://doi.org/10.7554/eLife.35439.015

## Data availability

All data generated or analysed during this study are included in the manuscript and supporting files.

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
