## [Decision Letter]

Thank you for submitting your article "A novel Cep120-dependent mechanism inhibits centriole maturation in quiescent cells" for consideration by *eLife*. Your article has been reviewed by two peer reviewers, and the evaluation has been overseen by a Reviewing Editor and Anna Akhmanova as the Senior Editor. The reviewers have opted to remain anonymous.

The reviewers have discussed the reviews with one another and the Reviewing Editor has drafted this decision to help you prepare a revised submission.

Summary:

The centrosome is the major microtubule-organizing center of animal cells and is comprised of a pair of microtubule-based structures termed centrioles surrounded by a pericentriolar protein matrix (PCM). In addition to forming the core of the centrosome, centrioles are also essential for the formation of cilia and flagella and play important roles in quiescent cells for cell signaling. Misregulation of centriole number and function has been linked with diseases such as cancer and neurodevelopmental disorders.

In this manuscript, Betleja and colleagues describe a novel role of CEP120 in maintaining centrosome PCM homeostasis during quiescence. They first established a quiescent cell assay to simultaneously induce G0 cell cycle arrest and deplete CEP120 by RNAi, thus preventing loss of centrioles. Subsequently, they found that depletion of Cep120 in quiescent cells led to preferential accumulation of PCM components (PCNT, Ninein and Cep170) on mother and daughter centrioles. Intriguingly, these elevated PCM negatively affected the organization of microtubule and disrupted the microtubule-based transport of centriolar satellites, resulting in defective cilia assembly and signalling. The authors propose a model in which CEP120 acts to suppress maturation of the daughter centriole in quiescent cells to enable efficient trafficking of proteins to the mother centriole to support cilia generation and function.

Overall, the reviewers felt that this is a well-executed study that describes an interesting role of CEP120. The results support most of the conclusions of the paper. However, all felt that the study is slightly light in providing the mechanisms by which CEP120 executes its function proposed by the authors. As a result, the study in its present form is somewhat preliminary to warrant publication in *eLife*. Nonetheless, the reviewers also felt that this issue can be addressed within the scope of *eLife*'s criteria for the revision. Therefore, we would like to invite you to submit a revised version.

Major points:

1) It would be useful to perform an RNAi add-back experiment to show the specificity of the CEP120 siRNA used throughout all the experiments. In doing so, we suggest to include the rescue experiments with deletion mutants as well. The authors can use deletion mutants that have been discussed previously in other studies, combine this with the analysis on the impact of recruitment of additional PCM components. Identifying some interesting separation of function mutants would be particularity insightful, especially if they can identify some specific domains, or better still, conserved amino acid sequences that are required to hold back PCM recruitment. If they could find a mutant of CEP120 that supports centriole duplication, but don't suppress PCM recruitment this would support a view of two distinct roles of CEP120.

Other essential revisions:

1) A key contention of this work is whether CEP120-dependent centrosome maturation also occurs in a non-quiescent state. In cycling cells, depletion of CEP120 indeed results in loss of centrioles. However, one could still detect the presence of mother centrioles with 48 hours CEP120 siRNA transfection. It will be interesting to test whether those mother centrioles also accumulate PCM components (Ninein and Cep170) upon CEP120 depletion. Another relatively simple way is to check if overexpression of PLK1 can rescue the PCM homeostasis defect in quiescent cells depleted of CEP120, which may completely bypass the requirement of CEP120, since PLK4 is a critical determinant of centriole maturation in nonquiescent cells.

2) The authors claim that depletion of CEP120 causes a centriolar satellite dispersal phenotype in quiescent cells. This conclusion is well supported by the imaging data in Figure 6D-F. However, I think it is still necessary for authors to examine the overall protein levels of those satellite components to firm up their conclusion.

3) Loss of Cep120 was found to cause a series of defects in PCM abundance and centriolar satellite localization during quiescence. How about the impact of CEP120 overexpression? The author may want to address this point and back up their CEP120 working model (Figure 6H) with gain of function in quiescent cells.

4) It would be helpful to quantify the level of CEP120 signal at the centrosome of control and CEP120 siRNA cells, rather than simply scoring whether cells are negative or positive for CEP120 (Figure 1D).

5) The authors report that while they observed an increase in various PCM components known to bind γ tubulin after knockdown of CEP120, they don't see a measurable increase in γ-tubulin itself (Figure 2I). This is surprising as microtubule nucleation is found to increase following CEP120 depletion. I think further exploration/discussion of this point is warranted.

6) The authors observed an increase in dynein recruitment to the daughter centriole following CEP120 depletion, but it is not clear to me why this would promote the dispersal of centriolar satellites.

7) A more detailed description of CEP120 antibody validation (both WB and IF) is needed, since this is such a critical tool in this study.

8) In Figure 4, the authors should provide details on how the "high" and "low" expression categories were defined.

9) In the Discussion, the authors write: "Because Cep120 loss did not completely mature the centrosome, it must be operating within a mechanism different than that of the Plk1-mediated maturation process." This is not necessarily true, as CEP120 may be just one target of Plk1 and as such it's depletion would not be expected to phenocopy Plk1 overexpression.

10) In the Discussion, the authors write: "Finally, our findings […] uncover a new type of molecular defect underlying the pathogenesis of ciliopathies such as Jeune and Joubert syndrome." In my view this is an overstatement, as the authors provide no evidence to show that the CEP120 mutations found in Jeune and Joubert patients influence the processes characterized in this manuscript.

---

## [Author Response]

Major points:1) It would be useful to perform an RNAi add-back experiment to show the specificity of the CEP120 siRNA used throughout all the experiments. In doing so, we suggest to include the rescue experiments with deletion mutants as well. The authors can use deletion mutants that have been discussed previously in other studies, combine this with the analysis on the impact of recruitment of additional PCM components. Identifying some interesting separation of function mutants would be particularity insightful, especially if they can identify some specific domains, or better still, conserved amino acid sequences that are required to hold back PCM recruitment. If they could find a mutant of CEP120 that supports centriole duplication, but don't suppress PCM recruitment this would support a view of two distinct roles of CEP120.

As noted in the manuscript, the specificity of the siRNA used in this study has been previously demonstrated (Comartin et al., 2013), which is why we chose it for these experiments. Nonetheless, we also verified the specificity of the siRNA in our assays as requested. To achieve this, we generated an expression plasmid expressing an siRNA-resistant version of Cep120, by making six silent mutations in the siRNA-recognition site (GFP-Cep120siRt; described in Methods). Transfection of GFP-Cep120siRt into Cep120-depleted cells rescued the centriole duplication defect in cycling cells (new Figure 2—figure supplement 1A). Importantly, it also rescued the PCM recruitment defects in quiescent cells (new Figure 2—figure supplement 1B-C). These new results are described in the subsection “Loss of Cep120 in quiescent cells causes accumulation of PCM” (second paragraph), which highlight the specificity of the siRNA used and add to the rigor of the analyses in the study.

To identify functionally important domains that are critical for the inhibitory function of Cep120, we tested various deletion mutants in a dominant-negative screen for PCM recruitment. As we showed in Figure 4A-B, overexpression of full-length Cep120 reduced PCM levels at the centrosome in a dose-dependent manner. This provides a relatively simple assay to test domains necessary for this inhibitory function. Using a series of deletion constructs that we previously established (Mahjoub et al., 2010), we overexpressed truncated versions of the protein and assessed the levels of PCM at the centrosome. Loss of the C-terminal coiled-coil domain required for targeting Cep120 to the centrosome failed to inhibit Pericentrin levels (new Figure 4C-D), indicating that localization to centrosomes is important for limiting PCM accumulation. Similarly, deletion of the N-terminal microtubule-binding domain abrogated Cep120 inhibitory function (new Figure 4C-D), supporting our hypothesis that binding to centriolar microtubules may be a mechanism for this inhibitory activity. Furthermore, deletion of the CPAP-interacting domain, shown to be necessary for centriole duplication (Lin et al., 2013) did not affect PCM levels. Collectively, these data suggest that both the microtubule-binding and centrosome targeting domains are essential for the inhibitory function of Cep120. These results are included in the subsection “Loss of Cep120 in quiescent cells causes accumulation of PCM” (last paragraph) and in Figure 4C-D. We thank the reviewers for the great suggestions.

Other essential revisions:1) A key contention of this work is whether CEP120-dependent centrosome maturation also occurs in a non-quiescent state. In cycling cells, depletion of CEP120 indeed results in loss of centrioles. However, one could still detect the presence of mother centrioles with 48 hours CEP120 siRNA transfection. It will be interesting to test whether those mother centrioles also accumulate PCM components (Ninein and Cep170) upon CEP120 depletion. Another relatively simple way is to check if overexpression of PLK1 can rescue the PCM homeostasis defect in quiescent cells depleted of CEP120, which may completely bypass the requirement of CEP120, since PLK4 is a critical determinant of centriole maturation in nonquiescent cells.

As noted in the Introduction section, the enrichment of Cep120 on the parental (daughter) centriole is relieved coincident with procentriole assembly in G1-S phase, a stage when the daughter centriole first matures into a mother centriole by definition. This process is distinct from the stage at which Plk1 activity regulates centriole maturation and PCM recruitment, which predominantly occurs at G2/M. This suggests that the mechanism(s) that regulate Cep120 association with parental centrioles is likely independent of Plk1, since those two processes are separated by the cell cycle. In support of this theory, a recent study demonstrated that over-expression of Plk1 in G1 can fully induce daughter centriole maturation without causing removal of Cep120 from the daughter centriole (Kong et al., 2014), again suggesting that these two pathways act independently of each other (we highlight this in the Discussion section). Thus, we were unsure of what exactly the reviewers were requesting with this specific experiment, particularly since they mention Plk1 and then Plk4 in the above sentence, two kinases that have very different functions. Nonetheless, we sought to determine whether cells containing the original parental daughter centriole showed enhanced PCM accumulation following loss of Cep120. We performed this analysis in non-serum starved cells in G1, since this is a stage when Cep120 is still enriched on the daughter centriole; in S-G2M stages Cep120 enrichment is lost from the daughter centriole (summarized in Figure 1A). We compared Pericentrin levels in cells treated with Sas6 siRNA (which still maintain Cep120 on the single daughter centriole) to cells transfected with Cep120 siRNA (which have lost Cep120 on the single daughter centriole). We noted a roughly 2-fold increase in Pericentrin levels on the single daughter centriole when Cep120 was absent (new Figure 2—figure supplement 2F), similar to the fold increase observed on the daughter centriole upon loss of Cep120 in quiescent cells (Figure 2B). This result is consistent with the hypothesis that the enrichment of Cep120 on parental centrioles helps to limit PCM recruitment, and that the removal of Cep120 at G1/S is likely mediating accumulation of PCM on that centriole.

2) The authors claim that depletion of CEP120 causes a centriolar satellite dispersal phenotype in quiescent cells. This conclusion is well supported by the imaging data in Figure 6D-F. However, I think it is still necessary for authors to examine the overall protein levels of those satellite components to firm up their conclusion.

We have quantified the overall levels of centriolar satellite proteins, using immunoblot analysis of wholecell lysates. We found that there is no change in the levels of PCM-1 and Ccdc11 (Figure 3). Similarly, there is no change in the level of the p150^Glued^ dynein-dynactin component that helps to traffic centriolar satellites. These results support our conclusion that the dispersal of satellites upon Cep120 loss is due to changes in localization of the cytosolic pool, and not changes in protein expression.

3) Loss of Cep120 was found to cause a series of defects in PCM abundance and centriolar satellite localization during quiescence. How about the impact of CEP120 overexpression? The author may want to address this point and back up their CEP120 working model (Figure 6H) with gain of function in quiescent cells.

As shown in Figure 4, overexpression of exogenous Cep120 caused a decrease in PCM abundance at the centrosome. Consistent with this observation, we now show that overexpression of exogenous Cep120 also results in dispersal of centriolar satellites. These data support our working model and are presented in new Figure 6G and in the Results section (subsection “Increased microtubule nucleation and dispersal of centriolar satellites upon Cep120 loss”, first paragraph). We thank the reviewer for the suggestion.

4) It would be helpful to quantify the level of CEP120 signal at the centrosome of control and CEP120 siRNA cells, rather than simply scoring whether cells are negative or positive for CEP120 (Figure 1D).

As recommended, we have quantified the level of Cep120 signal at the centrosome of control and Cep120-depleted cells. We found that Cep120 signal is almost completely abrogated, consistent with the quantification of the fraction of cells that contain Cep120. This new data is now included in Figure 1D.

5) The authors report that while they observed an increase in various PCM components known to bind γ tubulin after knockdown of CEP120, they don't see a measurable increase in γ-tubulin itself (Figure 2I). This is surprising as microtubule nucleation is found to increase following CEP120 depletion. I think further exploration/discussion of this point is warranted.

We were also somewhat surprised to see that γ-tubulin levels at the PCM did not increase upon loss of Cep120, even though we observed increased levels of microtubule nucleation from the centrosome (Figure 6A-C). However, it is now well established that the microtubule-nucleating functions of the γ -TURC can be modulated by changes in abundance (i.e. amount of protein complex at the centrosome), but additionally via enhanced activation. For example, PCM components such as Cdk5Rap2 have been shown to stimulate activation of the γ –TURC at the MTOC and cytoplasm (reviewed in Farache et al. Open Biol., 2018; Roostalu and Surrey, Nat Rev Mol Cell Biol., 2017). Therefore, the increased levels of γ –TURC activators (such as Cdk5Rap2) that we found at the centrosome may result in increased activation of resident γ –TURC, which would explain the increased microtubule nucleation we observed.

6) The authors observed an increase in dynein recruitment to the daughter centriole following CEP120 depletion, but it is not clear to me why this would promote the dispersal of centriolar satellites.

A number of studies have defined a critical role for the dynein-dynactin complex in establishing proper centriolar satellite organization, localization, and protein trafficking (reviewed in Hori and Toda, 2017 – cited in our manuscript). Since perturbing the localization/concentration of the dynein complex results in satellite dispersal phenotypes, one possibility is that the accumulated dynein at the centrosome means that there is less motor available in the cytoplasm to help recruit and traffic satellites. In support of this theory, we overexpressed Ninein, which recruits and binds to the dynein complex (specifically p150^Glued^; Mazo et al., 2016) to the centrosome.

Without depleting Cep120, increasing centrosomal Ninein alone resulted in dispersal of satellites (Figure 6K). However, we agree that this is simply a hypothesis, based on our interpretation of the results.

7) A more detailed description of CEP120 antibody validation (both WB and IF) is needed, since this is such a critical tool in this study.

We have included immunoblots of whole-cell lysates from MEF cells transfected with either control or Cep120-targeting siRNA. The antibody recognizes a predominant band of 120 kDa that is lost upon depletion of Cep120. We also probed whole-cell lysates of NIH3T3 fibroblasts and NIH3T3 cells stably expressing GFP-Cep120 (Mahjoub et al., 2010). The antibody also recognizes the exogenously expressed GFP-tagged protein. These data are presented in new Figure 1—figure supplement 1. Finally, the specificity of the antibody was shown by immunofluorescence multiple times throughout the manuscript (Figures 1-7): cells transfected with control siRNA oligo display the well-established asymmetric localization of Cep120 on daughter centrioles, as has been published by us and others using different anti-Cep120 antibodies (Mahjoub et al., 2010; Li et al., 2012; Comartin et al., 2013). Transfection with Cep120-targeting siRNA causes specific loss of this signal, which is unaffected by transfection with control or Sas6-targeting siRNA. Altogether, we believe that these data highlight the robustness and specificity of this antibody.

8) In Figure 4, the authors should provide details on how the "high" and "low" expression categories were defined.

In these experiments, we compared the ratio of exogenous and endogenous Cep120 protein at the centrosome by immunostaining cells with anti-Myc and anti-Cep120 antibodies. We consider 1-2X the relative amount of exogenous Cep120 (compared to endogenous protein) “low” overexpression, while “high” overexpression ranged from 2-3X. We have included this information in the legend of Figure 4A-B, as well as the Materials and methods section (subsection “Analysis of centrosomal and ciliary proteins:”, first paragraph).

9) In the Discussion, the authors write: "Because Cep120 loss did not completely mature the centrosome, it must be operating within a mechanism different than that of the Plk1-mediated maturation process." This is not necessarily true, as CEP120 may be just one target of Plk1 and as such it's depletion would not be expected to phenocopy Plk1 overexpression.

Indeed, this is another possible explanation and we thank the reviewers for highlighting it. We have modified the corresponding paragraph in the Discussion to include this alternative explanation (first paragraph).

10) In the Discussion, the authors write: "Finally, our findings […] uncover a new type of molecular defect underlying the pathogenesis of ciliopathies such as Jeune and Joubert syndrome." In my view this is an overstatement, as the authors provide no evidence to show that the CEP120 mutations found in Jeune and Joubert patients influence the processes characterized in this manuscript.

We agree with the reviewers that the sentence as currently constructed makes it appear that we are making this direct connection. We have tempered this statement as follows: “Finally, our findings establish the important and exquisitely sensitive control of centrosome PCM homeostasis and its relationship to primary cilium assembly and signaling, and uncover a *potentially* new type of molecular defect that *may* underlie the pathogenesis of ciliopathies such as Jeune and Joubert syndrome.”